# Impact of Subjective and Objective Factors on Subway Travel Behavior: Spatial Differentiation

**DOI:** 10.3390/ijerph192315858

**Published:** 2022-11-28

**Authors:** Qi Chen, Yibo Yan, Xu Zhang, Jian Chen

**Affiliations:** 1College of Architecture and Urban Planning, Chongqing Jiaotong University, Chongqing 400074, China; 2College of Civil Engineering, Henan University of Technology, Zhengzhou 450001, China; 3College of Traffic & Transportation, Chongqing Jiaotong University, Chongqing 400074, China; 4Jiangsu Province Collaborative Innovation Center of Modern Urban Traffic Technologies, Nanjing 211189, China

**Keywords:** built environment, subway travel behavior, subjective perception, spatial heterogeneity, multiscale geographically weighted regression

## Abstract

People’s perceptions and understanding of the built environment can shape and regulate travel intention and behavior. From the perspective of urban design, improving the built urban environment is an important way to encourage green travel. This study determined the impact path from the built environment to subway travel behavior, considering the intermediate effect of psychological factors. The impact path could provide feedback for optimizing the built environment, thereby improving the ratio of subway travel. In this study, the impact path hypothesis of “objective environment—subjective psychological—travel behavior” was first proposed, taking travelers’ psychological perceptions as the intermediary variable. Second, questionnaires and geographic information were used to measure the objective environment and subjective psychological perceptions. Third, a structural equation model was constructed to verify the proposed path hypothesis. Finally, multiscale geographically weighted regression was used to analyze the influence of subjective and objective factors on travel behavior and its spatial differences. The empirical case sampled 405 residents of Zhengzhou, China. The results verified the proposed impact path hypothesis and revealed spatial heterogeneity in its impact on travel behavior. The research explains how travel behavior is generated and could support the optimization of the urban built environment.

## 1. Introduction

At the 75th session of the United Nations General Assembly, China pledged to peak its carbon emissions by 2030 and achieve carbon neutrality by 2060. The issue of traffic and transportation, being significant emission sources, must be addressed to meet these goals [1]. As such, the transportation sector must develop low-carbon traffic facilities and promote green travel.

The *Green Travel Action Plan 2019–2022* was published by China’s Ministry of Transport in 2019, highlighting the need to optimize slow transportation services, refine related systems, and design better human-centric traffic and transportation spaces. The aim is to build a walkable, safe, and comfortable transportation system for urban living. Subsequently, the Ministry of Transport released the *Green Travel Development Action Plan* in July 2020, which prioritized green travel development and advocated a simple, low-carbon lifestyle. The public was encouraged to prioritize green travel methods such as walking or cycling; reducing the number of cars on the roads would also increase green travel in Chinese cities. Developing a low-carbon travel environment requires not only excellent urban designs but also scientific approaches to guide the public toward green travel and low-carbon lifestyles.

Guiding public transport travel by optimizing the design of the built environment has been widely recognized as important by scholars. Unlike the natural environment, the built environment is an artificial system that facilitates people’s activities in an urban space. People’s travel demands are generated by different land use patterns, and travel mode is determined by the distribution of facilities surrounding the resident. Additionally, travel decisions are affected by the urban space design, such as in terms of walking convenience and environmental comfort. For government departments to guide residents to take the subway, they must understand the processes through which the built environment affects travel behavior and thereby formulate strategies and plans to optimize the built environment.

Therefore, an examination of residential public transportation behavior (especially regarding subways) and psychological factors will provide a valuable reference for refining urban spatial and transportation designs that encourage residents to travel in a green manner by walking and taking the subway. The manuscript is organized as follows: first, extant research on the built environment and travel behavior are reviewed; second, the thought processes, methodology, and indicators associated with this study are described; third, empirical case data are analyzed, and the results presented. Finally, the findings of this study are discussed and conclusions are presented.

## 2. Literature Review

### 2.1. Mechanisms That Influence Travel Behavior

The built environment is material and human-constructed. It is associated with multiple elements, including land use, urban design, and road traffic. Current research on the quantitative calculations underlying the built environment is typically based on the 3Ds proposed by Cerveros in 1997: density (e.g., building and population density), diversity (e.g., functional diversity), and design (e.g., pedestrian path connectivity) [2]. As scholars researched the built environment in more depth, they added two more Ds—destination accessibility and distance to transit [3]. This indicator system for built environments was subsequently used in many research areas, including travel behavioral analyses.

By providing basic spaces and facilities for daily activities, the built environment is universally perceived to influence residents’ travel preferences regarding public transportation. Extant studies have focused on the correlation between the built environment and travel behavior, deriving some widely accepted conclusions such as the higher the building density, functional diversity, and level of public transportation convenience, the higher the ratio of green travel [4,5]. However, urban managers do not care about the correlation between the built environment and travel behavior, but rather their causal relationship and the psychological decision-making processes that influence travel behavior. Urban managers aim to optimize the built environment to induce changes in residents’ travel behavior.

Specific to this concern, psychological research frameworks and methodology play a vital role in understanding environmental issues and proposing solutions. Ajzen’s theory of planned behavior (TPB) was applied in myriad studies of the impact of psychological factors on individual travel behavior [6,7,8]. For instance, Chen surveyed residents in Chongqing and Chengdu and used structural equation modeling (SEM) for analysis. The results showed that attitudes, subjective norms, and perceived behavioral control have a significant positive impact on behavioral travel intentions [9]. Based on the TPB, Li introduced low-carbon transport policies as latent variables to construct a modified theory of planned behavior (MTPB) for the travel intentions of travelers expressed in 759 questionnaires collected from three cities. The results showed that the roles of low-carbon policies, subjective norms, and personal norms were most salient in the behavioral intention of adopting low-carbon travel modes [7]. The subjective perception model, as an extension of the TPB, achieved some consensus with its explanation of the psychological process underlying travel decision-making. However, insufficient consideration was given to the role of built environment elements, making it difficult to apply the study findings to urban design practices.

According to the theory of stimulus–organism–response (SOR), the process by which the objective environment influences travel can be described as follows: when people are stimulated by objective built environment elements they develop a subjective perception that is used to make judgments and travel decisions [10]. Therefore, scholars started considering the simultaneous impact of objective built-environment factors and subjective personal perception factors on travel behavior. The underlying idea was that objective factors influence subjective psychological factors, resulting in people making judgments and acting accordingly [11,12].

### 2.2. Mechanisms of Influence on Travel Behavior

To date, many empirical studies have revealed that travel behavior is influenced by road connectivity, subway station density, and other built environment variables, as well as individual attribute variables. Studies have also pinpointed relatively obvious spatial differentiation regarding the influence of some built environment variables on residential travel behavior [13,14,15]. Researchers of such spatial differentiation have suggested that different communities may have differing travel behavior due to urban spatial distribution and spatial heterogeneity among residential mobility behavior factors. As such, neglecting spatial influence may result in inaccurate study results [13]. Therefore, scholars introduced the geographically weighted regression (GWR) model to study the spatial heterogeneity of travel preferences.

Since there might also be differing extents of spatial heterogeneity of the various influential factors, scholars further developed the multiscale geographically weighted regression (MGWR) model based on the GWR model. The variable bandwidth in the MGWR model is subject to change, so adjustments can be made to the spatial heterogeneity of influence variables. This enhances the model’s explanatory power, allows for further examination of the impact of different extents of spatial heterogeneity, and expands the applicable scope of the GWR model in empirical research [16,17]. Along with the advancing maturity of the MGWR model, its scope has also extended to various fields, including travel behavior, disease prevention, and public health [18,19,20].

In travel behavior research, the MGWR model can be applied to urban regions, enabling researchers to utilize a spatial perspective to explore how residential travel behavioral factors are expressed across different cities [21]. In terms of the developmental processes underlying travel behavior, perceived objective factors develop a subjective understanding, affecting travel behavioral decisions. Based on the conclusions of extant studies, there is significant spatial differentiation in the influence of built environment factors on travel behavior. We can thus reasonably deduce that there may also be a level of spatial differentiation across residents from different regions within a city in terms of their perception of the built environment and subjective attitudes toward public transportation travel.

### 2.3. Summary

From the literature review, we established that correlation studies of the built environment and travel behavior are already relatively mature and that the psychological processes underlying transportation travel behavioral decisions have also been well explained. However, there may be differences in the subjective perception of non-objective environmental elements and attitudes regarding public transportation due to differing individual attributes and living environments. Therefore, it remains difficult for government agencies overseeing urban management to formulate a coherent and integrated strategy involving built environment elements, psychological perception processes, and public transportation travel behavior. This makes it challenging for them to optimize the built environment to guide changes in travel behavior.

To tackle this problem, this study first consolidates and reviews extant research on the impact of psychological and built environment factors on travel behavior. Building on the theoretical foundation that the built environment affects individuals’ subjective psychological perception and, in turn, travel behavior, this study also proposes that the residents’ decision to take the subway is simultaneously influenced by the built environment and psychological factors and that there is a significant spatial heterogeneity among these factors. This research empirically analyzes the study hypotheses. In technical terms, this study considers both the objective built environment and subjective psychological factors. Drawing on the theoretical foundation of the TPB, this study considers the characteristics of the built environment in the respondents’ residential area to validate the influence path of “objective environment–subjective perception–travel behavior”. Thereafter, the MGWR model is used to examine and estimate the impact of objective and subjective factors on subway travel behavior from the urban spatial perspective. Our study results could significantly contribute to low-carbon travel behavior theory and refine the low-carbon travel management system in city blocks. As such, this study provides a quantitative basis for travel policymaking and optimization of the urban built environment.

The innovations of this study are as follows: (1) Verifying the intermediate effect of psychological factors in the influence of the built environment on subway travel behavior; (2) Establishing the impact path of “built environment-psychological factors-behavioral intention”; (3) Exploring the spatial heterogeneity of effects on subway travel behavior.

## 3. Methodology

### 3.1. Overall Research Process

According to the TPB and various empirical studies, behavioral decisions are directly affected by subjective intentions, which are affected by subjective perceptions and attitudes. These are, in turn, affected by the objective environment, as highlighted by the SOR theory. Accordingly, the following was derived:

Thesis Statement 1: The objective and subjective factors underlying subway travel behavior are hypothesized to interact with each other. However, the objective built-environment factors are distributed unevenly across city spaces, which is concerning since the subjective attitudes toward subway travel are affected by such factors;

Thesis Statement 2: There is spatial heterogeneity in terms of the impact of the built environment and subjective perception factors on subway travel decisions.

The subjective latent variable was measured by surveying community residents on their subjective perceptions and attitudes. The objective environment latent variable was measured by obtaining geographical data on the residents’ objective subway travel environment. AMOS software was used to examine and test the SEM model hypotheses and calculate the influence path coefficient of the latent variables. After obtaining the influence path of the latent variables on subway travel behavior, the spatial heterogeneity hypothesis of subway travel factors was tested. The MGWR model derived the quantitative impact relationships for residential subway travel behavior. The overall research process is illustrated in Figure 1.

### 3.2. Influence Path Hypotheses of Subway Travel Behavior

First, using the TPB framework as reference, the influence path hypotheses (H1–H5) of the impact of subjective psychological factors on subway travel mobility were proposed, and Model A was developed. Drawing on Model A, environmental variables were introduced in the subsequent hypotheses (H6–H10), and Model B was developed. The specific model path hypotheses are as follows:
**Hypothesis** **1 (H1).***The level of acceptance of subway travel has a positive impact on subway travel intentions*;
**Hypothesis** **2 (H2).***The level of the travelers’ perceived societal stress has a positive impact on subway travel intention*;
**Hypothesis** **3 (H3).***The travelers’ control over their own behavior has a positive impact on subway travel intentions*;
**Hypothesis** **4 (H4).***The travelers’ control over their own behavior has a positive impact on subway travel behavior*;
**Hypothesis** **5 (H5).***The travelers’ subway behavioral intentions have a positive impact on subway travel behavior*.

Considering how the objective built-environment factors affect the residents’ subway travel behavior, it was proposed that travelers are first stimulated by the objective travel environment before forming a subjective understanding regarding subway travel. Based on this understanding, along with other subjective factors, travelers develop travel intentions, which in turn affect travel behavior. The following hypotheses were thus proposed:
**Hypothesis** **6 (H6).***A better subway travel environment has a positive impact on perceived behavioral control (i.e., travelers will be more inclined to travel by subway)*;
**Hypothesis** **7 (H7).***A better subway travel environment can increase the perception of social stress, encouraging them to choose subway travel*;
**Hypothesis** **8 (H8).***A better subway travel environment can affect the attitude of travelers when selecting subway travel, pushing them toward the idea of adopting this form of travel*;
**Hypothesis** **9 (H9).***A better subway travel environment has a positive impact on subway travel intentions*;
**Hypothesis** **10 (H10).***A better subway travel environment has a positive impact on subway travel behavior*.

The model path hypotheses are shown in Figure 2.

### 3.3. Measuring Variables

Based on the model path hypotheses, this study developed three subjective endogenous latent variables, two mobility behavior endogenous latent variables, and one objective environment exogenous latent variable. Three to five measurement variables were derived from each latent variable for quantitative measurements. It was ensured that all values obtained from the measurement variables had a clear indication regarding their impact on the latent variables. Each measurement variable is described in Table 1.

The subway travel environment variables were derived and identified using insights from extant research. Four measurement variables—subway station distance, level of functional diversity, road connectivity level, and distance from the city center—were selected and described (Table 2). Geographic data were obtained from the surrounding residential area. The built environment calculation methodology is as follows. The road connectivity variable reflects road connectivity and accessibility. Space syntax describes the spatial relations in the road network [22]. The level of functional diversity reflects the diversity of land utilization [23]; this is calculated using information entropy based on facility type [24]. Subway station distance refers to the distance of the nearest subway station from where the respondent lives. It reflects the accessibility of subway facilities [3]. The distance from the city center refers to the distance between where the respondent lives and landmark buildings in the city center. It reflects the regionality of where the resident lives and has been widely proven to be a significant factor in travel [25,26].

### 3.4. Spatial Heterogeneity Analysis of the Factors

A linear regression model was used to test the impact of objective and subjective factors on subway travel behavior. The quantitative relationship between the various factors and subway travel behavior was verified by obtaining factor scores of latent variables in the SEM as new variable scores and incorporating individual attribute factors. The MGWR model was then used to analyze the spatial differences in the impact of effective variables. The MGWR model is expressed as in formula (1).
(1)$$y_{i}=\sum_{j=1}^{k}\beta_{wj}(u_{i},v_{i})x_{ij}+\varepsilon_{i}$$
where $y_{i}$ represents the weighted value of the number i sample of the travel behavior latent variable; ($u_{i}$, $v_{i}$) represents the geographic marker of the number i sample; $u_{i}$ represents the longitude of the number i sample; and $v_{i}$ represents the latitude of the number i sample. $x_{ij}$ represents the control values for the post-weighted perceived behavioral control, travel behavior intention, and built environment latent variables; $\varepsilon_{i}$ represents random error; wj represents the bandwidth used for the number j variable regression coefficient; and $\beta_{wj}$ represents the regression coefficient of each variable.

## 4. Empirical Analysis

### 4.1. Study Data

This study focused on residents in the downtown area of Zhengzhou, Henan Province. Data were collected using random sampling. Zhengzhou is an important central city in the central part of China, serving as a national integrated traffic hub. There are approximately 6.5 million residents living in the downtown area. The city is currently in the rapid development phase of building its subway system, and the share of subway travel as a transportation mode is growing yearly. Data were obtained in two steps. First, a survey was conducted to collect data on the subjective attitudes toward subway travel behavior. Then, travel environment data were obtained by extracting geographic information from the respondents’ residential areas.

The psychological factor survey had two parts: the pre-survey and the official survey. Test samples were collected as part of the pre-survey, focusing on a small area. Based on the survey results, the questionnaire items were examined before the distribution, quantity, descriptions, and other aspects were adjusted for the final official survey. The official survey was conducted in December 2021 in public spaces such as large malls and parks with greater foot traffic. This ensured even spatial distribution in terms of the respondents’ residential neighborhoods. The on-site surveys incentivized participants by providing cash and presents. A total of 500 questionnaires were distributed. Excluding invalid samples such as those with incomplete responses, who failed the screening criteria, or who were not living in Zhengzhou, a total of 405 valid questionnaires were completed. The gender, age, and occupation distribution of valid respondents are shown in Table 3 below. The geographic information data of the travel environment were extracted in January 2022 using Bigemap software (http://www.bigemap.com/). Table 4 shows the statistical description of the objective variables.

The objective data on the subway travel environment were represented as vector geographic data sourced from Bigemap and Gaode Map. The data included various points of interest (POI), road networks, subway lines, and other Euclidean data. According to the residence location data obtained from the survey, 290 of the 405 samples were from the downtown area, encompassing most of the subway stations in the city center. The distribution of respondents and the subways are shown in Figure 3. The surrounding subway travel environment data of each respondent were quantified and extracted. These, along with the psychological factors data obtained from the survey, were entered into the SEM for analysis.

### 4.2. Data Reliability and Validity

Data reliability and validity of Model A and Model B were measured; the results are shown in Table 5. The latent variable reliability test indicator was Cronbach’s alpha. The test values were not lower than the recommended threshold of 0.6. This signifies that the data had good consistency. Confirmatory factor analysis (CFA) was conducted; the factor loadings were distributed between 0.6 and 0.9, with relatively high coefficients. The lowest value was above 0.4, signifying that the intention of the survey items was clear and had high differentiation. The lowest average variance extracted (AVE) was 0.36, and the highest was 0.61. This shows that the measurement model had acceptable explanatory power and that the latent variable met the convergence validity requirements.

### 4.3. Influence Path Testing of the Impact of Latent Variables Based on the SEM

Five indicators were used to assess the SEM: root mean square error of approximation (RMSEA), goodness of fit (GFI), comparative fit index (CFI), incremental fit index (IFI), and adjusted goodness of fit (AGFI). The results are shown in Table 6. Compared to the recommended reference values in the extant literature, the CFI and IFI values in Model A were lower than recommended, while the AGFI values in Model A and Model B were also lower than recommended. All other indicators met the requirements [27,28,29]. Comparing the test results of the two models, Model B’s various indicators were larger. Other than the AGFI being lower than the recommended value, all other indicators met the requirements for model fit. The standardized path coefficients of the various latent variables in both models are shown in Table 7. The hypotheses (H1–H5), derived from the TPB, demonstrated a strong statistical significance in both models (*p* < 0.01). Therefore, the hypothesis paths were valid. From the influence path of the built environment latent variables, the hypotheses regarding the impact of built environmental factors on psychological ones (H6–H8) and travel behavior (H10) were valid (*p* < 0.01). However, the hypothesis regarding the built environmental impact on travel intention (H9) was not valid (*p* > 0.05). The valid influence paths for Model A and Model B hypotheses are shown in Figure 4 and Figure 5. Thesis Statement 1 proposed in this study was validated.

### 4.4. Spatial Heterogeneity Analysis Based on MGWR

Personal income, age, education, and other variables are important factors that affect psychological states and travel behavior [30,31]. Regarding the varying occupation characteristics and model requirements for the type of dummy variables, this study recorded the occupation variable. Unemployed individuals and students were classified as unemployed; white-collar and blue-collar workers, as well as service personnel, were classified as employed. Retirees were classified as retired. The results showed that other than attitudes, subjective norms, perceived behavioral control, and subway travel environment, individual attribute variables, including age, education, and occupation, had a significant impact on subway travel. Table 8 provides additional details.

According to the linear regression results, the model verified the overall effectiveness of the dependent variable. However, the linear regression model did not consider the spatial changes in terms of the variable relationships. Excluding variables that were not statistically significant, six variables were retained: attitudes, perceived behavioral control, subway travel environment, travel behavior intentions, age, and education. These were then incorporated into the GWR and MGWR models. The model bandwidth used was based on the self-adaptation method of the Gaussian kernel function. The model evaluation indicator was the corrected Akaike information criterion (AICc). The lower the AICc score, the higher the explanatory power of the model [17]. Table 9 illustrates that the R^2^ and AICc values of the MGWR model were better than those of the multiple linear regression and GWR models. For the MGWR model, the AICc and residual sum of squares were smaller than those of the multiple linear regression and GWR models. Therefore, the MGWR model had superior explanatory power. In Table 9, VAR denotes variables, VAL indicates valid, and DIF represents difference. The VAL indicator expresses the ratio of samples that exhibited statistical significance with a 95% confidence interval. The DIF indicator in the MGWR model reflects spatial differences in the variables. A higher DIF value represents a greater spatial difference in the impact on travel behavior.

Based on the spatial regression results shown in Table 9, the impact of all six indicators exhibited spatial heterogeneity. As such, thesis statement 2 was valid. The MGWR’s variable spatial coefficients are distributed, as shown in Figure 6. This shows that the attitude factors in the western region, as well as the impact of the subway travel environment on travel behavior in the eastern region, were relatively low in significance. All other indicators demonstrated statistical significance in all regions.

## 5. Discussion

Urban policy managers are gradually paying more attention to policies and urban renewal strategies that encourage green travel by combining walking and public transportation. Previous empirical studies have demonstrated a significant correlation between changes in the built environment and travel behavior. However, such a correlation does not indicate a cause-and-effect relationship. When the objective is to guide changes in travel behavior, such a correlation is difficult to apply to urban renewal strategies. Therefore, the mechanism by which the built environment affects subway travel behavior is critical in facilitating green travel in practice. There were two main perspectives in this study: (1) The objective and subjective factors of subway travel behavior were hypothesized to interact with each other; (2) The proposal of spatial heterogeneity concerning the impact of the objective and subjective factors on subway travel decisions.

This study considered an empirical case study of Zhengzhou to validate the above thesis statements. The influence path of “objective built environment-subjective psychological perception-actual travel behavior” was validated. From the spatial analytical perspective, this study described the impact of objective and subjective factors on subway travel behavior. For ease of comparing results, this study developed Model A and Model B (shown in Figure 4 and Figure 5). Model A is a travel behavior model developed based on the basic framework of the TPB. The subjective psychological factors were analyzed as exogenous latent variables. Table 7 shows that this influence path hypothesis was valid, demonstrating that psychological factors affected travel behavior decisions. As with previous travel behavior studies, this is a valid conclusion, but it still failed to establish a cause-and-effect relationship between the built environment and behavior [8,32,33]. Building on Model A, Model B incorporated subjective and environmental factors as exogenous latent variables, expanded the basic framework of the TPB, and established the influence path of “environment-perception-behavior”. The results showed that this path can be used effectively to examine subway travel behavior and is consistent with another study on bicycle travel behavior. These two studies also validated that the objective environment affects travel behavior regarding the road network, accessibility, state of facilities, and other elements [11].

According to the results of the SEM (as shown in Table 7), the following discussion is presented: (1) Based on the impact paths of the environment TE-ATT-TB and TE-SN-TB, we found that the built environment can influence people’s attitudes and perceptions of social pressure on subway travel, which in turn affects people’s travel behavior. In addition, according to Table 5, the factor loadings of distance to the city center and functional diversity both exceeded 0.7; these were the main variables measured among the built environment latent variables. We speculate that due to the developed subway system and high density in the central area of Zhengzhou, the ratio of subway travel is higher there than in other areas; (2) The impact path of TE–PBC–TB shows that a good environment for subway travel can reduce the perceived difficulties of subway travel. This suggests that improved walkability and perception of subway stations could guide people to choose subway travel; (3) Hypothesis H9, referring to the path between TE and TBI, was not supported based on the SEM results. It can be inferred that the impact of TE on TBI could be explained by the intermediate effect of subjective psychological factors, which is also supported by other studies [34,35]; (4) The impact path of TE-TB is clear and has been confirmed by many empirical studies [13,14]. This indicates that people do not simply choose a travel method based on their subjective attitudes but also are restricted by their objective environment.

According to the spatial heterogeneity analysis, the impact of psychological factors and built environment factors on subway travel behavior exhibited spatial heterogeneity in Zhengzhou due to the spatial distribution of built environment elements in the urban space. Spatial heterogeneity is an important issue that cannot be ignored when formulating urban renewal strategies to promote people’s choice of metro travel. The spatial heterogeneity analysis in Table 9 showed that the impact of ATT had the most significant spatial heterogeneity. TE and PBC also showed strong spatial heterogeneity, which validates Thesis Statement 2. As shown in Figure 6, ATT did not have a statistically significant impact on travel behavior in the western region of Zhengzhou, where land use is predominately newly developed, and the population density and infrastructure remain at low levels. One speculation is that due to the low development density and incomplete infrastructure, the respondents did not have an accurate perception of the subway travel environment, and their attitude toward subway travel was highly subjective and subject to considerable uncertainty. Additionally, the impact of TE in the eastern region was also not statistically significant. Here, we propose an explanation based on the current situation in the eastern part of Zhengzhou. That is, due to the more developed economy and infrastructure, people living in the eastern part of Zhengzhou have a variety of choices of travel mode, and the impact of the built environment on travel mode choice is inconsistent.

Other than objective environmental and subjective attitude factors, we also found that age and education had a significant impact on subway travel decisions. It can also be seen in Figure 6 that the coefficients for age and educational attainment did not vary spatially in Zhengzhou City; thus, it can be assumed that these variables are global, i.e., individual attribute factors do not show significant spatial heterogeneity due to differences in urban distribution. From the results in Table 8, we also gathered some interesting insights; that age and subway travel were significantly and strongly correlated and that the older one is, the less likely one chooses to travel by subway. However, from the regression results on the occupation factor, retirement status demonstrates a near-significant correlation with subway travel. This diverges from the conclusion regarding the impact of age. That said, Table 3 indicates that older adults aged above 55 years only represented 7.1% of the sample data, while retirees comprised 12.8%. This shows that some of the respondents had already retired even before reaching retirement age. Based on this finding, we conjecture that as age increases, the income and social status of the employed also increases, resulting in them being more likely to prefer traveling by car, which they find more comfortable. However, once they have retired, they prefer to travel by subway. Additionally, education also exhibited a strong correlation with subway travel behavior. The results in Table 9 show that those with lower educational qualifications preferred to travel by subway and that the impact of education on subway travel behavior was rather consistent across the city. This also shows that those with higher educational qualifications are more rational and autonomous in decision-making and are less likely to be affected by external factors.

According to the results, some suggestions that may help encourage subway travel are proposed. (1) Among the three latent variables of psychological factors, subjective norms have a significantly larger influence on travel behavior than attitudes and perceptual behavioral control. This suggests that the perceived pressure of low-carbon travel can be increased through social media and newspapers to promote people’s choice of subway travel. (2) The built environment has a significant effect on all three psychological factors, and the built environment can directly influence people’s travel behavior. Moreover, the factors of distance to the city center and functional mix had a higher weight in the latent variable of TE. This suggests that strengthening the development of urban sub-centers and enriching the diversity of land use could help to increase the share of subway travel. (3) In different areas of the city, some built environment elements had different impacts on subway travel behavior due to differences in the level of economics and infrastructure. In some cases, improving the built environment cannot effectively shift travel choices from private cars to public transport. This suggests that improving the subway travel environment in places with weak infrastructure could help encourage people to take the subway.

## 6. Conclusions

This study used structural equation modeling to validate the influence path hypotheses of “objective environment-subjective attitude-travel behavior” and demonstrated that subjective perception and attitudes play a mediating role in the impact of the built environment on travel behavior. MGWR was also used to validate the spatial heterogeneity of the impact that objective and subjective factors have on travel behavior. The study results further uncovered the decision-making processes underlying subway travel, from which a complete feedback mechanism was developed to help optimize the built environment to guide changes in travel behavior. This can serve as a point of reference for public agencies when they develop policies and embark on urban renewal. The main conclusions from this study are as follows: (1) We verified the intermediate effect of psychological factors under the influence of built environment on subway travel behavior; (2) We established the impact path of “built environment-psychological factors-behavioral intention”; (3) We verified the spatial heterogeneity of the impact of built environment and psychological factors on subway travel behavior.

Based on the models’ explanatory power, the level of our analysis, and perspectives on how to optimize built environments and develop policies that encourage green travel, we believe that future research should include profiling residents and further segmenting their characteristics to explore the travel preference of different types of residents. This could result in more targeted and effective strategies for elevating urban renewal. A limitation of this study is that simple sampling was used to collect data without considering individual characteristics. Additionally, the study also used the traditional on-site approach of collating data via questionnaires. This required the respondents to answer many questions within a short period. This may have resulted in some responses being affected by emotions and the environment. Therefore, the survey methodology could incorporate wearable devices in the future to reduce survey errors caused by subjective preferences and improve data reliability. Finally, the study did not survey the residents’ travel objectives. This may have limited the influence mechanism in explaining subway travel behavior. The influence path hypotheses and indicator system could have considered the impact of travel purposes.

## Figures and Tables

**Figure 1 ijerph-19-15858-f001:**
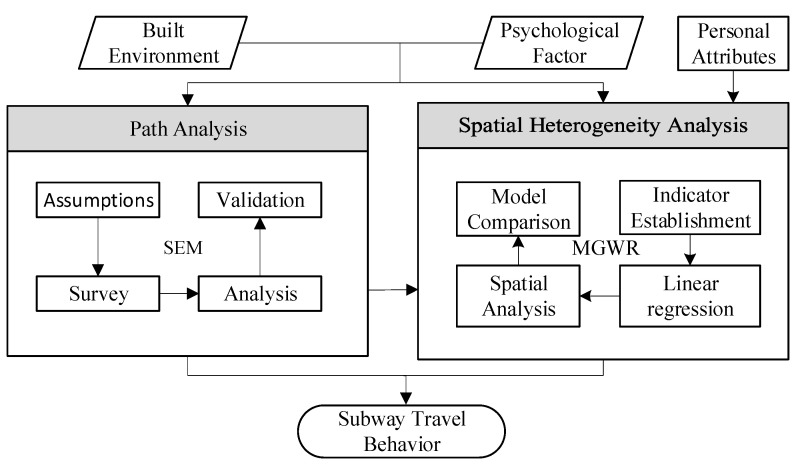
Model development process.

**Figure 2 ijerph-19-15858-f002:**
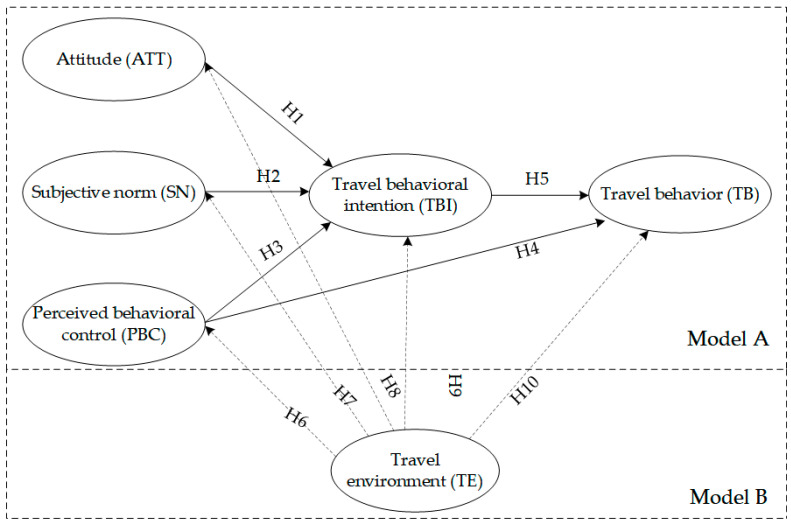
Model path hypotheses.

**Figure 3 ijerph-19-15858-f003:**
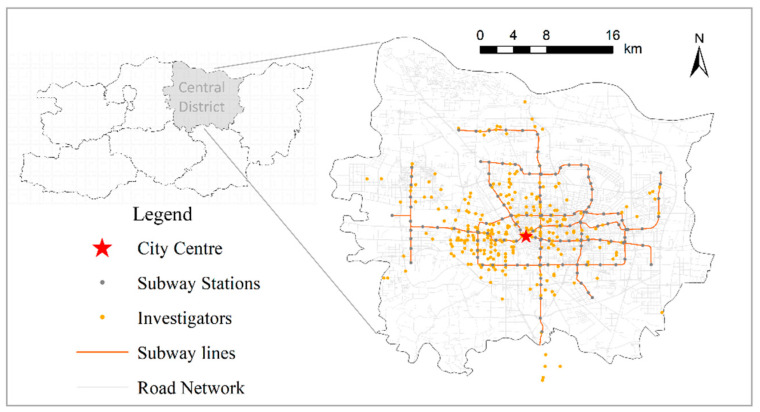
Spatial distribution of respondents.

**Figure 4 ijerph-19-15858-f004:**
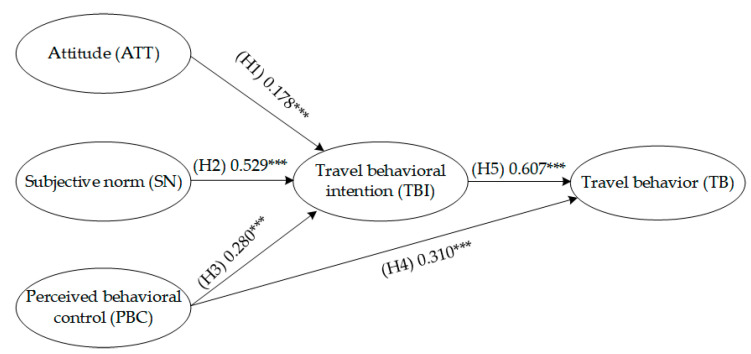
Model A latent variables influence paths; *** *p* < 0.01.

**Figure 5 ijerph-19-15858-f005:**
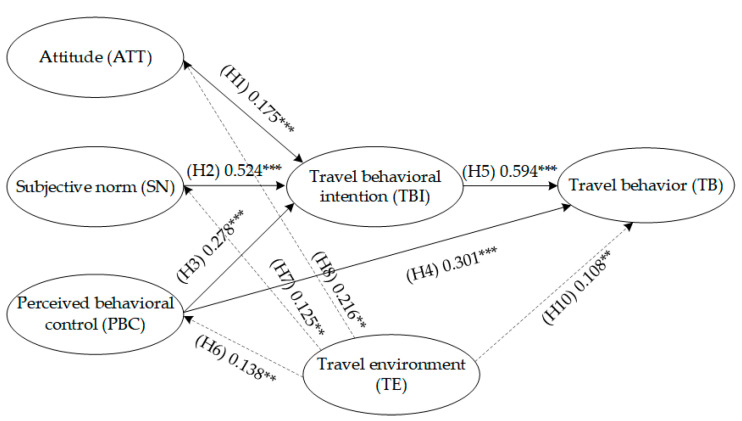
Model B latent variables influence paths; *** *p* < 0.01, ** *p* < 0.05.

**Figure 6 ijerph-19-15858-f006:**
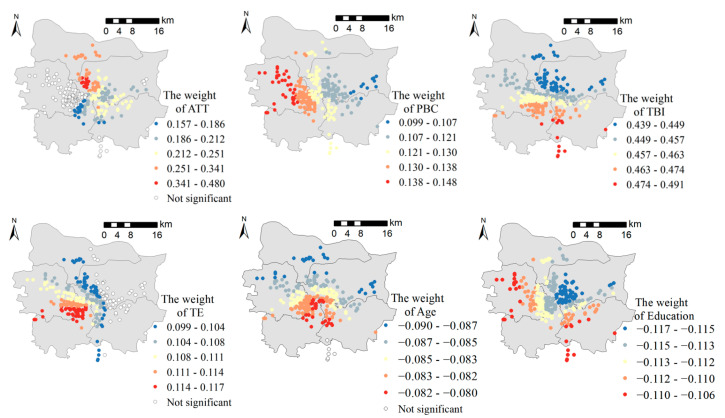
Spatial distribution of the impact of various factors.

**Table 1 ijerph-19-15858-t001:** Measurement variables.

Latent Variable	Measurement Variable	Code	Explanation
Attitude (ATT)	Subway travel can significantly relieve traffic jams.	ATT1	An evaluation of whether a behavior is positive or negative
	Taking the subway is very safe.	ATT2	
	Subway travel is convenient.	ATT3	
Subjective norms (SN)	I choose to travel by subway to reduce urban environmental pollution.	SN1	Societal stress derived from engaging in a certain act
	I choose to travel by subway to reduce traffic jams.	SN2	
	I choose to travel by subway because my family and friends often recommend me to do so.	SN3	
Perceived behavioral control (PBC)	I am very satisfied with the environment while waiting for the train inside the subway station.	PBC1	Level of difficulty in completing an action
	I can find the subway station and entrance very easily.	PBC2	
	I do not find the subway crowded when traveling.	PBC3	
Travel environment (TE)	Road connectivity level	TE1	Quantification of the travel environment when choosing to travel by subway
	Level of functional diversity of facilities	TE2	
	Subway station distance	TE3	
	Distance from city center	TE4	
Travel behavior intentions (TBI)	I am happy to choose subway travel.	TBI1	Travel intentions of respondents
	I will travel more by subway in the future to reduce environmental pollution	TBI2	
	I will travel more by subway in the future to reduce traffic jams.	TBI3	
Travel behavior (TB)	Subway travel is my main mode of commuting.	TB1	Travel behavior of respondents
	Subway travel is my main travel mode when I am not commuting	TB2	
	For everyday travel, if I can take the subway, I will not choose to travel using my own car	TB3	

**Table 2 ijerph-19-15858-t002:** Subway travel environment measurement variables.

Selected Indicator	Schematic Diagram	Indicator Description	Calculation Method
Road connectivity level	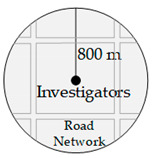	Space syntax was used to describe three indicators: the road network’s holistic concentration, control value, and connectivity. The indicators’ information entropy was used to describe the road network’s spatial relations, as well as calculate and obtain the road connectivity variable.	$E_{i}=-ln{\left(n\right)}^{-1}=\overset{n}{\underset{j=1}{\sum }}p_{ij}ln\;p_{ij}$ $Q_{i}=\frac{E_{i}}{k\times 1-\sum_{i=1}^{n}E_{i}}$
Level of functional diversity	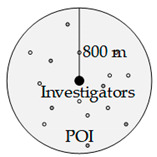	Information entropy was used to measure functional diversity. The functions of buildings were classified accordingly into eight types: dining, enterprise, leisure, school, hospital, government, skyscraper, and shopping. Information entropy is then used to calculate the level of functional diversity in the area.	
Subway station distance	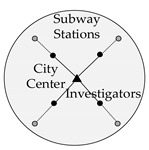	Linear distance between subway stations near the residential neighborhood.	Measured by spatial distance.
Distance from city center		Linear distance between the neighborhood and the city center.	

**Table 3 ijerph-19-15858-t003:** Statistical description of individual attribute variables.

Statistical Variable	Classification	Number of Samples	Ratio
Gender	Male	196	48.40%
	Female	209	51.60%
Age	≤18	10	2.47%
	18–30	250	61.73%
	30–40	80	19.75%
	40–55	36	8.89%
	55–65	12	2.96%
	65 and above	17	4.20%
Occupation	White-collar worker	38	9.38%
	Commerce and service	136	33.58%
	Blue-collar worker	98	24.20%
	Unemployed	27	6.67%
	Student	54	13.33%
	Retired	52	12.84%
Education	Primary school	2	0.49%
	Junior high school	20	4.94%
	Senior high school	57	14.07%
	Bachelor	281	69.38%
	Master and doctor	45	11.11%
Personal annual income (monetary unit, RMB. Average level in 2022 was 96,400)	Below 50,000	187	46.17%
	50,000–100,000	123	30.37%
	100,000–150,000	56	13.83%
	150,000–250,000	19	4.69%
	200,000–250,000	7	1.73%
	250,000 and above	13	3.21%

**Table 4 ijerph-19-15858-t004:** Statistical description of travel environment variables.

Travel Environment Variable	Average	Max	Min	SE
Distance from subway station (m)	1041.69	6602.13	10.24	992.08
Distance from city center (m)	6405.90	17,474.56	253.65	4034.21
Road connectivity index	8.11	21.34	0.90	5.22
Functional mix	0.20	0.75	0.00	0.14

**Table 5 ijerph-19-15858-t005:** Measurement variables.

Measurement Variable	Model A			Model B		
	CFA	AVE	Cronbach’s	CFA	AVE	Cronbach’s
			Alpha			Alpha
ATT1	0.57	0.36	0.60	0.57	0.36	0.60
ATT2	0.66			0.65		
ATT3	0.55			0.56		
SN1	0.89	0.61	0.77	0.89	0.61	0.77
SN2	0.89			0.89		
SN3	0.50			0.50		
PBC1	0.72	0.39	0.61	0.72	0.39	0.61
PBC2	0.63			0.63		
PBC3	0.51			0.51		
TBI1	0.50	0.58	0.78	0.50	0.58	0.78
TBI2	0.88			0.88		
TBI3	0.85			0.85		
TB1	0.58	0.39	0.67	0.58	0.39	0.67
TB2	0.56			0.57		
TB3	0.71			0.72		
TE1	—	—	—	0.70	0.52	0.79
TE2	—			0.86		
TE3	—			0.46		
TE4	—			0.79		

**Table 6 ijerph-19-15858-t006:** Model fit test results.

Test Indicator	RMSEA	CFI	GFI	AGFI	IFI
Reference value	<0.10	>0.85	>0.85	>0.85	>0.85
Model A	0.10	0.82	0.86	0.80	0.82
Model B	0.08	0.85	0.87	0.83	0.85

**Table 7 ijerph-19-15858-t007:** SEM path coefficients and hypotheses testing results.

Path Hypothesis	Model A		Model B		
	Path Coefficient	*p*-Value	Path Coefficient	*p*-Value	Hypothesis Verification
ATT—>TBI	0.18	0.00	0.18	0.01	H1 is valid
SN—>TBI	0.53	0.00	0.52	0.00 ^(2)^	H2 is valid
PBC—>TBI	0.28	0.00	0.28	0.00	H3 is valid
PBC—>TB	0.31	0.00	0.30	0.05 ^(1)^	H4 is valid
TBI—>TB	0.61	0.00	0.59	0.00	H5 is valid
TE—>PBC	—	—	0.14	0.04	H6 is valid
TE—>SN	—	—	0.13	0.03	H7 is valid
TE—>ATT	—	—	0.22	0.00	H8 is valid
TE—>TBI	—	—	0.01	0.87	H9 is not valid
TE—>TB	—	—	0.11	0.00	H10 is valid

Note: ^(1)^ The results are significant at *p* ≤ 0.05 (within the 95% confidence interval); ^(2)^ the results are significant at *p* ≤ 0.01 (within the 99% confidence interval).

**Table 8 ijerph-19-15858-t008:** Linear regression model.

Variable	Standardized Coefficient	*p*-Value	Standard Error	VIF
ATT	0.15	0.00	0.05	1.69
SN	−0.04	0.43	0.05	1.86
PBC	0.15	0.00	0.05	1.37
TE	0.09	0.02 ^(1)^	0.17	1.11
TBI	0.49	0.00 ^(2)^	0.05	1.76
Male	−0.01	0.80	0.16	1.09
Age	−0.10	0.03	0.09	1.41
Education	−0.10	0.01	0.12	1.14
Financial situation	−0.01	0.91	0.07	1.34
Unemployed	0.07	0.08	0.22	1.26
Retired	0.09	0.06	0.28	1.50
Adj-R2	0.44			
AICc	929.95			

Note: The dependent variable is subway travel behavior. ^(1)^ The results are significant at *p* ≤ 0.05 (within a 95% confidence interval); ^(2)^ the results are significant at *p* ≤ 0.01 (within a 99% confidence interval).

**Table 9 ijerph-19-15858-t009:** Geographically weighted regression results.

	GWR				MGWR					
VAR	Mean	Min	Max	VAL	BD ^(1)^	Mean	Min	Max	DIF ^(2)^	VAL
ATT	0.16	0.12	0.19	100%	148	0.15	−0.10	0.48	121%	53%
PBC	0.14	0.11	0.16	100%	397	0.13	0.10	0.15	33%	100%
TE	0.10	0.06	0.12	88%	404	0.11	0.09	0.12	26%	83%
TBI	0.46	0.45	0.51	100%	387	0.46	0.44	0.49	11%	100%
Age	−0.09	−0.09	−0.07	85%	404	−0.08	−0.09	−0.08	20%	98%
Edu	−0.12	−0.12	−0.10	100%	404	−0.11	−0.12	−0.11	10%	100%
Adj-R^2^		0.44					0.47			
AICc		930.08					921.77			

Note: The dependent variable is subway travel behavior. ^(1)^ BD refers to the bandwidth. ^(2)^ $\mathrm{DIF}=\frac{\left(max-min\right)}{\left|max\right|}\cdot 100\%$.

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
