# Peer review of "Impact of Subjective and Objective Factors on Subway Travel Behavior: Spatial Differentiation"

_ijerph, 2022, doi:10.3390/ijerph192315858_

Round 1
Reviewer 1 Report
The paper gives valuable information supported by data. The overall level of the paper is good. However, I have just a few small comments on the manuscript.
More attention should be devoted to presenting and discussing the results. Further discussion of the results is necessary. Please present a more detailed discussion of the results for increasing the availability and reliability of the study and understanding the results. This helps the reader to clearly understand the contribution of the study. Also, this would strengthen the conclusions of the paper.
Conclusions need to be strengthened. Please, improve it, add some key results supported with data, and make it more effective.
Based on the key findings and conclusions, it may be better to give more recommendations for future analyses.
Reviewer 2 Report
The paper presents good investigation results and useful for understanding the impact of subjective and objective factors on subway travel behavior. After some editorial changes and further explanation, the paper is worth to be published.
1. The abstract should also include essential aspects of the study methods.
2. Title of Table 9 should be provided.
3. The index of Figure 6 should be made appear sharper and clearer.
4. The paper should be carefully reviewed again and revised since some editing errors are found.
Reviewer 3 Report
Article: Impact of Subjective and Objective Factors on Subway Travel Behavior: Spatial Differentiation
The issues of the article are very topical in relation to urban transport. Many cities around the world are taking steps to encourage travellers to use public transport.
Below are my comments and questions:
- In the Abstract and Introduction, it is worth adding the purpose of the study,
- In the Introduction, it is also worth adding (defining) the Subjective Factors and Objective Factors that the Authors include in the research.
- The literature review is very modest. Particularly with regard to section 1.1 line 70, where there is a reference to psychological factors (1991 item). Here the literature should be supplemented and contemporary items should be added.
- H4 and H5 are the same:
H4: The travellers' control over their own behaviour has a positive impact on subway (line 186) travel behaviour. (line 187)
H5: The travelers' control over their own behaviour has a positive impact on subway (line 188) travel behaviour (line 189).
- What guided the authors in setting the sample size at 500 respondents?
- Conducting the survey in purposively selected locations is not random, it is purposive (it concerns pedestrian traffic).
- The research would have been more complete if the Authors had also included travellers on the metro. Why did they not do so?
- The Discussion is worth supplementing by referring to the results of research by other researchers on a similar problem in other cities. Do they confirm the findings of the research presented in the article, or are they different?
Reviewer 4 Report
The Paper may be considered after minor revision. Comments are mentioned in the annotated file.
